# We know what we need: Older adults' and stakeholders' perspectives on ageing, health, and wellbeing in Pakistan

Sonia Sameen[1]*, Javeria Mansoor[1], Maryam Pyar Ali Lakhdir[1,2], Muhammad Asim[1], Rose Munyoki[1], Carolyn Greig[3,4], Justine Davies[5,6], Bilal Ahmed Usmani[1]*

1 Department of Community Health Sciences, Aga Khan University, Karachi, Pakistan, 2 Institute of Health Policy, Management and Evaluation, Dalla Lana School of Public Health, University of Toronto, Canada, 3 School of Sport, Exercise and Rehabilitation Sciences, University of Birmingham, Birmingham, United Kingdom, 4 MRC-Versus Arthritis Centre for Musculoskeletal Ageing Research, University of Birmingham, United Kingdom, 5 Institute of Applied Health Research, University of Birmingham, Birmingham, United Kingdom, 6 Department of Global Health, Centre for Global Surgery, Stellenbosch University, South Africa

* bilal.usmani@aku.edu (BAU), sonia.sameen@aku.edu (SS)

## Abstract

The rapid increase in the older adult population worldwide, especially in low- and middle-income countries (LMICs) like Pakistan, raises critical challenges for healthcare, social support, and policy planning. Older adults in LMICs, especially in culturally diverse regions like Sindh, Pakistan, often face limited access to healthcare, economic insecurity, and social isolation. These challenges highlight the need to understand ageing in this context and explore potential solutions at multiple levels of the socio-ecological model. This qualitative descriptive study used focus group discussions (FGDs) to gather information from 24 older adults and 24 stakeholders in urban Karachi, and 26 older adults and 26 stakeholders in rural Thatta. Each FGD consisted of 4–6 participants. They were conducted using a semi-structured guide, moderated in local languages, and analyzed thematically using the socio-ecological model as a guiding framework. At the individual level, mental health concerns like loneliness, depression, and stress were prevalent, with participants emphasizing the importance of family and community support. Technological barriers were noted, particularly in urban areas where older adults expressed a desire to learn to use smartphones and social media. At the sociocultural level, participants highlighted a lack of respect, social integration, and the need for senior care homes. They noted that family support was crucial but also expressed the need for socializing opportunities and greater societal recognition. At the structural level, difficulties accessing healthcare and managing medications were major concerns. Additionally, there was a lack of awareness about existing social security programs which could be expanded to support older adults. The study emphasizes the complexity of ageing-related issues in LMICs and the importance of addressing personal,

**Data availability statement:** All relevant data supporting the findings of this study are available within the manuscript and its Supporting Information files. Specifically, anonymized qualitative summaries, including the condensed thematic analysis (S2 File) and the anonymized themes and quotes table (S3 File), have been provided as supplementary material. The complete transcripts and full coding framework contain potentially identifying information and therefore cannot be shared publicly due to ethical restrictions imposed by the Aga Khan University Ethics Review Committee. Researchers may request access to these materials from the corresponding author or through the Aga Khan University Ethics Review Committee, subject to institutional approval. Please get in touch with the AKU ERC using this email address: erc.pakistan@aku.edu.

**Funding:** This study is a substudy of a broader research project originally supported by the Institute of Global Innovation, University of Birmingham, UK, for which the award was received by Professor Justine Davies and Professor Carolyn Greig. The original study is available at DOI: https://doi.org/10.1371/journal.pone.0304474. The data used in this analysis were collected as part of that original study. The current substudy did not receive any additional or dedicated funding. The funders had no role in the study design, data collection and analysis, decision to publish, or preparation of the manuscript.

**Competing interests:** The authors have declared that no competing interests exist.

sociocultural, and structural barriers. These study findings may contribute to discussions around improving support systems for older adults in similar contexts and fostering a more inclusive approach to ageing in resource-deprived settings, like Pakistan.

## Introduction

The growing number of older adults in the population necessitates greater attention to the health and well-being of this demographic. Recognizing vital needs related to ageing is essential for strategic planning to support the wellness and welfare of older adults [1]. In Pakistan, the older population was approximately 12.5 million in 2017, with projections estimating it could reach 40 million, or 8% of the population, by 2050. Globally, projections indicate that by 2050, one in six people will be 65 years or older, reflecting a significant demographic shift driven by declining fertility rates and increasing life expectancy [2,3]. Physiological changes accompanying ageing increase the risk of chronic conditions, a challenge strongly felt in low- and middle-income countries (LMICs) [2]. Many older adults also face various challenges, including financial instability post-retirement, often resulting in dependency on family members or caregivers [4]. These challenges, however, cannot be understood in isolation as they are shaped by the broader social determinants of health that influence how people age and the opportunities available to them.

Social determinants of health, i.e., the conditions in which people are born, grow, live, work, and age, significantly shape health outcomes. These factors include socioeconomic status, education, neighborhood, employment, social support networks, and access to healthcare. Research suggests that social determinants can impact health more significantly than healthcare or lifestyle choices, accounting for 30–55% of health outcomes [5]. Addressing these social determinants is crucial for improving health and reducing disparities [6]. Understanding how older adults and those who support them define and interpret healthy ageing, i.e., maintaining functional ability and well-being in older age, within these social contexts is therefore an important first step.

In Pakistan, distinct differences exist between life in rural and urban areas. These contextual differences not only determine access to resources but also create distinct forms of vulnerability. Poverty rates are highest in rural regions [7], and a substantial portion of the older population lives in poverty [8]. Conversely, older adults in urban centres typically have better access to nutritious food, clean water, and electricity [9]. Rural older adults often face material deprivation [10], whereas urban counterparts may experience social isolation and weakened community networks [11]. Understanding these contrasts is crucial to developing context-sensitive approaches to healthy ageing.

Most studies in Pakistan have focused on urban settings, exploring older adult needs, experiences, and challenges. Commonly reported issues include financial problems, social needs such as social interaction and religious involvement,

limited employment opportunities, and loneliness [12,13]. As a LMIC, Pakistan faces disparities at both individual and sociocultural levels that influence healthy ageing [14]. These structural and contextual disparities often translate into barriers that limit older adults' ability to age healthily and maintain autonomy [15]. Although few previous studies exist about the objective health outcomes and needs of older adults [8], those studies were quantitative and therefore do not fully capture ageing individuals' lived experiences, perceptions, and priorities that shape well-being.

Given the strong influence of social determinants on autonomy, dignity, and quality of life [16], there remains a critical gap in understanding how older adults themselves interpret and navigate these factors within the Pakistani context. A qualitative approach is essential to explore how older adults perceive these determinants and navigate challenges related to ageing, healthcare access, and social inclusion. Collectively, this evidence not only contextualize quantitative findings but also sheds light on hidden barriers and enablers that shape well-being. Exploring what supports or resources enable older adults to remain active and engaged can reveal important facilitators of healthy ageing and inform the development of more person-centered interventions.

Neglecting older adult health and social needs can lead to increased healthcare costs and decreased quality of life for individuals and families, creating inter-generational impacts as younger family members take on care-giving roles. At the policy level, despite the introduction of national policy for older adult health in 1999, Pakistan is yet to implement it. Additionally, very few institutions in the country recognize geriatric care as a specialty with general practitioners and other specialists typically providing healthcare services to older adults [17]. These gaps highlight systemic weaknesses in Pakistan's approach to geriatric care. Addressing these systemic gaps requires culturally grounded, context-specific solutions that reflect urban–rural disparities and promote equitable, dignified ageing through both policy and community-level action. Building on this evidence, this study aimed to understand ageing in Pakistan by examining how urban–rural differences shape older adult and stakeholder experiences, challenges, and opportunities for healthy ageing, and to inform policies that promote equitable and dignified ageing. It sought to capture perspectives from both older adults and key stakeholders to better understand how these determinants shape experiences of ageing. The study was guided by four overarching research questions: [1] how do older adults and stakeholders define healthy ageing; [2] what barriers hinder the achievement of healthy ageing; [3] what facilitators or enablers support it; and [4] what solutions or recommendations can strengthen the well-being of older adults across diverse contexts.

## Methods

### Study design

This qualitative descriptive study was embedded within a broader multi-method research project titled "Exploring healthy ageing trajectories and determinants in Pakistan: a mixed-methods study," [16] which investigated the priorities and needs of the older adult population in Pakistan. The main study aimed to identify the needs and priorities of older adults and stakeholders across urban and rural locations, exploring their perspectives on ageing, key areas of importance, available services, and barriers to living well in later life. In the quantitative phase, participants ranked their health and social priorities and needs through a structured voting process. The same participants were then invited to participate in the qualitative follow-up study to explore their perspectives on barriers, proposed solutions, and recommendations for social determinants of health and well-being. The Consolidated Criteria for Reporting Qualitative Research (COREQ) checklist was used to ensure effective reporting of the various aspects of this study. [16,18]

### Study setting, population, and recruitment

Two districts in Sindh province were purposively selected for data collection: District East, Karachi, representing an urban setting, and District Thatta, representing a rural setting. Although both sites are in the same province, they exhibit notable geosocial differences. Karachi, the largest city in Pakistan and the provincial capital attracts people from across the

country with various religious backgrounds for career opportunities, with an estimated population of 20 million (2023). In contrast, Thatta has a population of around 1 million, primarily Sindhi-speaking, with Muslim and Hindu communities forming the majority [19].

The study population comprised two major groups:

1) Older adults, aged sixty years and above, were recruited with the assistance of a community mobilizer and a research coordinator. The community mobilizer, a trusted local representative familiar with the communities, reached out to potential participants through household visits, word-of-mouth communication, and coordination with local community leaders and neighborhood committees. Information about the study was shared verbally in local languages to ensure clarity and accessibility. The research coordinator maintained contact with those who expressed interest, confirmed eligibility based on inclusion criteria, and scheduled participants for workshops. Referrals among participants were also used to identify additional individuals willing to participate.

2) Stakeholders were also invited to participate in the study, including a diverse group of individuals and organization partners known to support older adults. These included: members of senior councils, representatives from the Ministry of Local Government and Ministry of Health, Relevant NGOs and private organizations (such as HANDS (Health and Nutrition Development Society), Darul Sukoon, Edhi), local authorities, community leaders, health and social care managers and informal caregivers including family members involved in elderly care. All stakeholders were based in Karachi and Thatta and were known to have experience working with or supporting older adults. Invitations were sent via email, as well as in-person visits by the study coordinators.

The call for recruitment was advertised on September 25th, 2021, for both urban (Karachi) and rural (Thatta) study centres. Recruitment continued from September 26th, 2021, to September 30th, 2021, and workshops were organized on October 1st and 6th, 2021, in Thatta. Another study coordinator helped recruit participants from Karachi from October 7th, 2021, to October 12th, 2021. Workshops in Karachi were organized on October 13th and 14th, 2021.

## Data collection

We conducted ten focus group discussions (FGDs) with older adults and stakeholders in Karachi and Thatta, holding five FGDs separately with older adults and stakeholders in each location. There were 24 older adults and 24 stakeholders in Karachi, and 26 older adults and 26 stakeholders in Thatta, with each focus group consisting of 4–6 participants. A quota sampling technique was used, with the study team recruiting a fixed number of older adults from the target populations for convenience. For details about the recruitment process, refer to the main study.

The semi-structured guide was developed using literature and expert opinion [20]. The study teams in both Karachi and Thatta undertook training on qualitative data collection procedures. The study team from Aga Khan University moderated in-person FGDs. Discussions lasted 1.5 to 2 hours in Karachi and 2 to 2.5 hours in Thatta. Separate focus groups were conducted for older adults, with gender-segregated groups in the rural site (Thatta) to respect cultural norms. Translators fluent in local languages, i.e., Sindhi, were present to ensure that all perspectives were accurately understood.

The data collection was aimed at achieving data saturation. The discussions were audio-recorded, transcribed, and subsequently translated into English. An overview of the data collection and analysis process is presented in Fig 1.

## Ethics statement

Ethical approval was obtained from the Aga Khan University Departmental and Institutional Review Committee (2021-6147-17944). Access to the Thatta site was granted after obtaining permission from the local management. All participants voluntarily participated and provided written consent. No identifiable participant information was recorded, and confidentiality and privacy were maintained throughout the study.

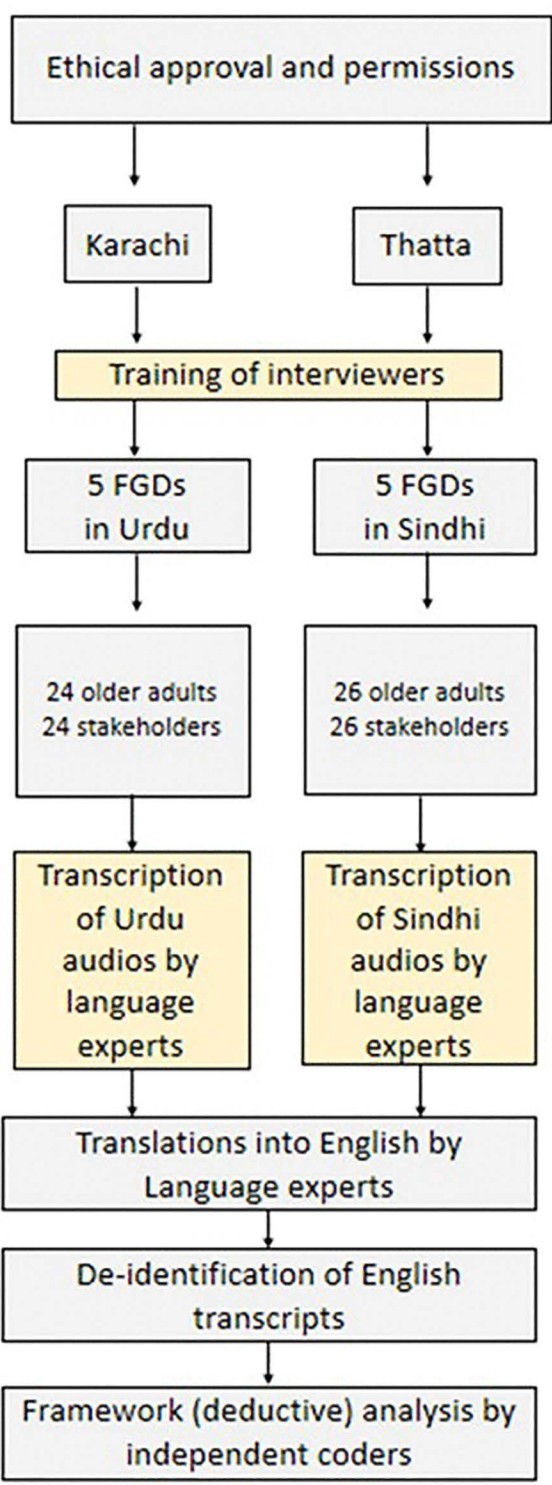

**Fig 1. Data collection and analysis workflow for the qualitative study conducted in urban Karachi and rural Thatta, Pakistan.**

## Data analysis

Thematic deductive analysis was conducted using the socio-ecological model framework, which consists of themes at three levels, i.e., Individual, Sociocultural, and Structural [21]. At the individual level, it focuses on personal characteristics such as knowledge, attitudes, skills, and biological aspects. The sociocultural level emphasizes the role of relationships, community norms, and social networks in influencing actions. Lastly, the structural level examines the broader systemic and environmental factors, such as policies, economic systems, and organizational structures, that create the context in which individuals and communities operate. Together, these levels highlight the interconnectedness of personal, social, and structural elements in shaping societal outcomes [22].

The FGDs were guided by a semi-structured discussion guide (S1 File), which was developed around the study objectives and aligned with the three levels of the framework. The data was manually coded by one researcher using an inductive-deductive thematic approach following Braun and Clark [23]. In the initial step, the researcher first familiarized themselves with the data and identified key elements related to specific research questions. This process informed the development of preliminary codes, which were subsequently refined, consolidated, and organized into groups reflecting similar ideas aligned with the study's main research questions. Codes were grouped into broader categories and themes reflecting barriers and solutions to healthy ageing across individual, social, and structural levels. This inductive analysis was conducted to generate initial themes after which these themes were subsequently mapped deductively onto the three levels of the socio-ecological model. All analyses and data management were conducted using Microsoft Excel (Microsoft Corporation, Redmond, WA, USA). Patterns were compared across stakeholder groups and geographical regions to explore differences and similarities.

## Researchers' reflexivity

The research team comprised eight public health professionals, three with formal training in qualitative research and four with experience in ageing-related studies. The primary researcher (female), with a background in public health and ageing, led the study design, data analysis, and interpretation, while senior qualitative researchers (one male and two females) provided oversight and methodological guidance. Their prior work with older adults and community-based health programs may have shaped their sensitivity toward issues of ageing, care, and social connectedness.

To reduce potential bias, FGDs were conducted by trained community members fluent in local languages and familiar with local customs. Their involvement helped build trust and encouraged participants to share their views openly. The lead researcher developed the coding framework, which was first thoroughly reviewed by a male senior qualitative researcher from Pakistan with contextual knowledge, and subsequently by two female senior researchers to enhance analytical rigor.

The authors recognize that their professional backgrounds in public health and ageing research may have influenced how the data were interpreted. Regular reflexive discussions and the use of the participants' own words helped ensure that findings accurately represented participant perspectives and strengthened the study's credibility.

## Results

The mean age of older adults was 67.3 years in Karachi and 67.2 years in Thatta, while the mean age of stakeholders was 31 years in Karachi and 36 years in Thatta, respectively. This study identified perceived barriers and proposed solutions for older adults in our population, categorized into three levels: individual, sociocultural, and structural (refer to Table 1).

At the individual level, major concerns included physical weakness, stress, loneliness, and technology barriers. At the sociocultural level, primary concerns were the need for family support and affection, respect and dignity, and a lack of socialization opportunities. At the structural level, concerns centered around the need for government support, limited access to health services and essential medicines, financial instability, and inadequate infrastructure and public services.

 

**Table 1. Summary of key barriers and corresponding solutions to promote healthy ageing across multiple levels of influence.**

| Level of Framework | Barriers | Solutions |
|---|---|---|
| Individual Level | • Generalized weakness leads to low motivation, increased susceptibility to diseases, and diminished willpower<br>• Loneliness, dependency, and fear of death affect mental and emotional well-being<br>• Technological barriers hinder adaptation to modern tools, though older adults express a strong desire for digital literacy. | • A sense of importance and acknowledgment enhances emotional well-being.<br>• Gratitude and religious practices provide emotional relief and promote well-being<br>• Early preparation both mentally and financially helps overcome ageing-related barriers and achieve personal goals. |
| Sociocultural Level | • Lack of family support, both emotionally and practically, negatively impacts well-being.<br>• Lack of respect and empathetic listening diminishes emotional contentment.<br>• Limited opportunities for socialization hinder social engagement and satisfaction. | • Family support and trust enable delegation of responsibilities, promoting self-care and better nutrition.<br>• Spending quality time with family fosters emotional well-being and reduces frustration.<br>• Strengthening community networks and creating dedicated spaces for older adults enhance social engagement and support. |
| Structural Level | • Lack of government support and insufficient pension systems undermine financial security.<br>• Limited access to affordable healthcare and medical supplies challenges health and well-being.<br>• Poor infrastructure, pollution, and lack of public facilities negatively affect health. | • Improved access to healthcare and essential medicines is vital to meet older adult needs.<br>• Subsidies and priority services for senior citizens are crucial for financial and social support.<br>• Ensuring access to basic utilities and financial stability improves quality of life. |

The proposed solutions at the individual level included fostering a sense of purpose, practicing gratitude, and engaging in religious practices. At the sociocultural level, recommended strategies involved encouraging family support through quality time, shared responsibilities, and joining community networks.

### Individual level

This level includes barriers and proposed solutions that address individual concerns. The barriers are categorized into three sub-themes. Proposed solutions include fostering a sense of purpose, practicing gratitude, engaging in religious prayers, and preparing for early ageing.

#### A. Barriers

**Generalized weakness leads to low motivation, increased susceptibility to diseases and diminished willpower.** Older adults in urban and rural Pakistan noted that generalized weakness was a major concern. They added that this weakness, in turn, resulted in other issues like irritability, being in bed throughout the day, and low motivation to perform physical activity. An older adult from Karachi expressed his feelings as follows:

*Sometimes, with age (age > 60), a person feels physically weak, and you feel like resting, nothing else I want to say. – FGD3 OK4*

Rural participants also mentioned that apart from weakness, their bodies had become prone to developing certain diseases, experienced low immunity, and had a higher risk of falls. A participant said:

*As people age, they experience weakness and become more susceptible to disease. Their weakness worsens gradually; they lose courage and willpower with age. – FGD3 OT1*

**Loneliness, dependency, and fear of death affect the mental health and emotional well-being of older adults.** The participants from Karachi and Thatta mentioned that loneliness resulted from losing their partners or siblings,

and children becoming busy in their careers and personal lives. Another reason for mental health issues highlighted was a reduced sense of power due to a lack of employment and, subsequently, money. A few stakeholders also mentioned sleep disturbance in older adults, resulting in an improper routine. This becomes a challenge for the family that looks after them.

*Loneliness is because of the death of a partner, or children getting busy with their jobs or personal lives. Dependency on children rises to the extent that they depend on their children at that age. Certain factors like memory loss and children support are included in dependency– FGD1 SK1.*

Another stakeholder from the urban site mentioned that older adults tend to become agitated due to perceived loneliness, which happens as they feel that their children do not listen to them.

*They show more anger and throw tantrums; they become hyper easily, and nobody listens to them. To make themselves noticed, they ignore others as well. So, according to me, in old age, a person gets angry more frequently, their children do not listen to them, and their children leave them in old age homes. – FGD3 SK4*

Some older adults also reported distress because they were unable to visit their loved ones, primarily due to weakness. A female from the rural site said:

*Whenever there is a wedding, a funeral, or I wish to go to pay my respects and condolences to someone, I cannot even go there. Relatives, dear ones, strangers, I cannot go to meet anyone. – FGD4 OT4*

Many older adults mentioned that they often dwell on the loss of friends and family and experience fear regarding their deaths.

*The problems and tensions have increased; relatives have left us. Our blood (relatives) left us, youthfulness left us, and slowly, everyone just left us like that. Fear of death is also there. Someone's mother dies or someone's father dies and we do not only have children and family ahead of us, but there is also only a 'Hereafter' ahead of us. Many hardworking people have departed. – FGD5 OT2*

**Technological barriers hinder older adults' ability to adapt to modern tools, but they express a strong desire for digital literacy.** Older adults, particularly those from urban areas, discussed difficulties adapting to recent technologies and expressed interest in learning how to use smartphones and explore the internet. They mentioned facing both technological and educational barriers. An older adult stated:

*Then technology and education-related barriers; even the children who are going to school know about it, my grandsons are doing it, they tell me how to use a computer, just show me how to do it, and they teach me. – FGD2 OK3*

They also expressed a desire to use social media platforms to stay engaged. Another older adult added:

*Independence in using technology is needed; then we would be able to use Facebook, play games, chat, or watch YouTube throughout the day. – FGD3 OK1*

## B. Solutions

**A sense of importance and acknowledgment enhances the emotional well-being of older adults.** Both older adults and stakeholders mentioned that older adults need to accept themselves as an essential part of the community, keep themselves busy, avoid overthinking, develop a proper sleep schedule and routine, and care for themselves.

*First of all, we should forgive others. Take rest, take care of our diet, and take medicines on time. You should also make friends. – FGD3 OK3*

The stakeholders reported that giving attention, time, and importance to older adults makes them feel respected and content.

*The first thing is if we give importance to our elders' opinions, they will stay happy but if we ignore them or do not give importance to their opinions, they will get hurt. It will speed up their ageing process more. – FGD2 SK4*

**Gratitude and religious practices provide emotional relief and promote well-being among older adults.** Both older adults and stakeholders reported that practicing gratitude and prayers helped alleviate mental health issues such as stress, anger, and loneliness. They found these self-care activities beneficial. Some participants also mentioned that reading newspapers and watching positive content on television contributed to their happiness.

*Gradually, you find yourself focusing more on your faith, your grandchildren, and your own well-being, distancing yourself from the fast-paced life you once led. – FGD1 OK1*

**Early preparation, both mentally and financially, helps overcome ageing-related barriers and achieve personal goals.** Most older adults suggested that preparing for healthy ageing should start early, encompassing mental preparation and financial planning. This insight was more profound among older adults of Karachi. One older adult shared:

*We need to mentally prepare ourselves, so that we can get it right. If we want to do something, we can—ageing is not a barrier. A person should mentally prepare for what he wants to achieve, and then it becomes possible. – FGD1 OK2*

### Sociocultural

This level also includes themes related to low support from family, respect and dignity, and avenues to socialize. The solutions proposed by the participants were categorized into themes such as delegation of responsibilities to family, spending time with family, and socializing.

### A. Barriers

**Lack of family support, both emotionally and practically, significantly impacts the well-being of older adults.** Nearly all participants (stakeholders as well as older adults) from both sites emphasized that family support is a top priority for older adults. They noted that retirement and physical limitations make older adults increasingly dependent on their children to buy groceries and medicines. Additionally, older adults feel a need for emotional connection, wanting to share their feelings and life experiences with their children and grandchildren. Participants also expressed that a peaceful home environment is essential for their well-being. Some mentioned that their children only join them for dinner on holidays, and a few participants highlighted concerns about the attitudes of their children and in-laws.

*If a person's children are spoiled and uncaring, it adds further stress and worry to their mind. However, if the child is obedient, the older person can find some encouragement and support during times of illness or conflict. – FGD4 OT1*

*Sometimes, when a daughter-in-law joins the family, if she is kind and seeks to keep everyone together, the home becomes a peaceful place. However, if she causes conflict, it can disrupt the family's harmony. – FGD5 OK1*

Stakeholders echoed these concerns, emphasizing that children must take responsibility for their older adult parents. One stakeholder stated:

*Children should serve their elderly parents, particularly by ensuring they have a healthy diet. If the parents are unwell, their children should make arrangements for their treatment. They should fulfill their responsibilities and make every effort to ensure their parents remain healthy and well. – FGD4 ST1*

**Respect and empathetic listening contribute to older adults' emotional well-being and contentment.** The participants, especially the stakeholders, collectively noted that showing respect to older adults enhances their mood and sense of contentment. They added that empathetic listening is crucial and complained that young people nowadays are often eager to voice their arguments first. Some stakeholders also mentioned that family members playing loud music disrupts older adults, especially during prayer times.

**Limited opportunities for socialization hinder older adults' social engagement and well-being.** The participants mentioned that very few opportunities were available for them to socialize. They said they do not have access to any community or recreational centres, especially designated for older adults. A participant from the urban site said:

*There should be opportunities for socializing and a pleasant environment, with trees and greenery, so that when you look out the window, at least, you feel good. – KF1 OK1*

## B. Solutions

**Family support and trust enable older adults to delegate responsibilities, ensuring they focus on self-care and proper nutrition.** Both older adults and stakeholders suggested that supportive family dynamics, where stakeholders build trust with their older adults, help them delegate responsibilities, allowing them to focus on self-care. A few participants also mentioned that family members need to take responsibility for providing a specialized diet for older adults, considering their capacity to chew solids and their dietary requirements. One older adult reported:

*You need a separate diet, but since your family members will not cook separate dishes for you, you have no other choice but to eat tea and rusk. – FGD5 OK1*

**Spending quality time with family fosters emotional well-being and reduces frustration in older adults.** Both stakeholder and older adult participants from urban and rural sites pointed out that spending time with family and creating peaceful, loving surroundings were essential for their emotional well-being. It also made them release their frustrations and share their life experiences. A stakeholder said:

*If you sit and listen to their life stories then they will be easy-going and joyful, but if you do not then they become very cranky and annoyed. – FGD5 SK2*

**Strengthening community networks and creating dedicated spaces for older adults enhance social engagement and support.** The stakeholders from both sites especially emphasized that socializing and building stronger community support networks are valuable. The neighbors, local community, and volunteers are essential for strengthening community support networks.

*Neighbors should take care of their old neighbors. On a community level, an NGO should be established that will serve to help old people financially. – FGD4 ST1*

Moreover, establishing institutions for senior citizens and community parks to provide spaces for engagement and recreation was pointed out as a favorable solution, especially when their children are not available at home during their working hours.

*The community should provide smaller clubs for senior citizens and everybody where you can sit and gossip. – FGD3 OK3*

*The government should establish institutions for people aged 60 years and above, where only people who have retired will pool in their ideas on the basis of their experiences so that they can also become valued citizens. – FGD4 ST1*

**Structural**

This component involved most concerns at the state policies and governance levels. The themes related to concerns were government support, access to health services, medication support, financial stability, and infrastructure. The identified themes within the proposed solutions included the provision of medicines, senior citizen subsidies, the provision of basic facilities, and improved access to health facilities.

**A. Barriers**

**Lack of government support and insufficient pension systems affect the financial security of older adults.** Many participants, especially those from rural areas, expressed concerns about their eagerness to continue working but were unable to do so. Older adults who were still working complained that their daily wages or salaries were delayed for months, emphasizing the need for a pension system. They pointed out that although a pension system exists, it is limited to retired government employees. Given that a large proportion of the population works in the private sector or is self-employed, these individuals are not eligible to benefit from the government pension system when they grow old.

*If the government cannot help them financially, then they should provide small jobs to the elderly so they can work and earn. – FGD2 SK1*

Limited access to affordable healthcare and necessary medical supplies challenges the health and well-being of older adults.
All the participants raised concerns that medical services were expensive, and many older adults develop conditions like diabetes and hypertension and require a regular supply of medicines. A few participants also demanded the provision of wheelchairs and ambulances, especially for older adults.

*The absence of a medical clinic in our village is a source of stress for us. There is a medical store. – FGD4 ST2*

**Poor infrastructure, including pollution and lack of public facilities, negatively impacts the health of older adults.** The participants, predominantly from Karachi, complained about air pollution and the lack of clean green parks and roads. Few participants from the rural site raised concerns about public toilets.

*In our country, the government is not doing anything. There is too much dirt over here, which is hazardous to our health. – FGD3 OK1*

**B. Solutions**

**Enhanced access to healthcare and medicines is essential for addressing the needs of older adults.** A few urban stakeholders mentioned that online services should be utilized in this regard, including online delivery of medical supplies or online mental health counselling services, e.g., Tele-clinics for older adults.

*(We) need a proper healthcare system for that, from where you can buy free medicines, the government should give us free medication and free check-ups. – FGD3 OK3*

**Subsidies and priority services for senior citizens are crucial for financial and social support.** Many participants, particularly from Karachi, compared the senior citizen benefits, including ration cards, discounts on bus fares, and priority service while waiting in lines at pharmacies, banks, or any other public place where token system tickets are being issued with their settings. They mentioned that these services are available in our neighboring countries and proposed that such schemes should also be available to them.

*We want our organization to provide us with financial aid. We do not have enough strength to make money on our own. We need external support. – FGD4 OT4*

**Access to basic utilities and financial stability are essential for improving the quality of life for older adults.** Most participants, especially from the rural site, demanded proper provision of facilities like electricity, gas, and water from the government, while some urban participants demanded some aid with groceries.

*In terms of facilities, they are the same as those that I mentioned earlier, like the availability of electricity, water, and gas. Other than this, having a stable and good income is important. – FGD4OT1*

These findings informed the development of an integrated framework illustrating urban–rural disparities and potential pathways for promoting healthy ageing in Pakistan (Fig 2).

## Discussion

Through the FGDs at an urban and rural site of Sindh province in Pakistan, the study explored older adult and stakeholder priorities regarding barriers and solutions at three levels of the socio-ecological model. Healthy ageing has become a primary area of focus globally with the increasing older adult population [24]. Being a LMIC, the study's overall concerns about urban and rural older adults were similar and were primarily directed toward providing basic amenities.

At the individual level, one of the identified themes included mental health concerns, loneliness, and stress. In community-dwelling older adults across Asia, factors such as having a spouse or partner, living with family, maintaining a large social network, frequent contact with friends and family, and receiving emotional and practical family support are all associated with fewer depressive symptoms and greater life satisfaction [25]. Some older adults in the study also reported that these reasons were associated with depression. A 2022 systematic review by Lapane et al. on older adults in congregate long-term care settings found that loneliness was prevalent among residents, even though they lived with other older adults. This loneliness was associated with increased risks of depression and suicidal tendencies [26]. A 2023 scoping review by Mazumder et al. on mental health interventions found that older adults in South Asia face significant disparities in mental health. While depression and anxiety were the most commonly reported issues, other outcomes included poor quality of life, cognitive decline, low self-esteem, and reduced physical capability. The review proposed solutions such as meditation, behavioural and educational interventions, technology-assisted approaches, and music therapy [18].

Participants also noted a technology barrier at the individual level, particularly in urban areas, where many older adults expressed an interest in learning to use smartphones and social media applications. A recent systematic review on Mobile Health (mHealth) use among older adults also found limited technological capacity in this group, highlighting that motivation and social support are essential for sustained engagement. However, none of the studies in this review focused on older adults from LMICs [27]. A Canadian study on older adult technology use similarly reported smartphones as the preferred device, but, in contrast to the study's findings, it found that most older adults could self-learn technology [28] without relying on their children or grandchildren.

A study from the UAE explored the rights of older adults in Islam and their country and reported that Islam supports providing shelters and senior homes for older adults without successors or relatives. These homes should be constructed

## Integrated Framework for Addressing Urban–Rural Disparities in Healthy Ageing in Pakistan


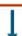

**Fig 2. Integrated Framework for Addressing Urban–Rural Disparities in Healthy Ageing in Pakistan.** The framework highlights distinct and shared barriers, mechanisms for change, and tailored solutions that bridge urban–rural gaps to promote equitable, inclusive, and dignified ageing.

based on Islamic programs, services, and foundations. Neglect of older adults is considered agonizing, and it contributes to the worsening of their health and psychological problems [29]. Pakistan is also an Islamic country, and the laws and sociocultural norms are mainly based on Islamic teachings. Many older adults in the study noted that their sense of contentment increased as they practiced religion more at this stage of life. However, they also highlighted the scarcity of senior homes at the sociocultural and structural level, proposing that such facilities should be established. This finding aligns with a study from Bangladesh, which highlighted the need for senior homes and reported that financial constraints often contribute to the hardships the older adult population faces. To address this, the study suggested expanding and equitably distributing social security programs [30]. Interestingly, the participants in this study did not mention any existing social security programs.

A study by Chantakeeree et al. conducted in Thailand emphasized the importance of older adults taking personal responsibility for their health, through consistent medication adherence, healthier lifestyle choices, and cultivating

spirituality to enhance emotional resilience [31]. While this focus on individual agency is important, it must be understood within the broader structural context. For instance, a study from Michigan, USA, revealed that many older adults struggled with external determinants of health, such as the affordability of healthcare, especially mental health services, concerns about access to safe water, and dissatisfaction with local infrastructure [32]. These findings note that personal responsibility, though vital, is often shaped and constrained by systemic conditions, highlighting the need for an enabling environment to support older adults in managing their health effectively. Similarly, this study also found that self-reliance was often proposed as a more sustainable solution compared to dependence on government support. In line with this, Parra et al. explored older adult views on physical activity and mindfulness, showing that mindfulness workshops enhanced self-awareness, nurtured self-compassion, and strengthened social connections. These findings align with the results of this study, where participants emphasized the importance of preparing for healthy ageing in advance [33].

Many older adults in this study, particularly from Thatta, expressed a strong desire to work despite their advanced age, motivated primarily by economic need but also by the wish to engage in meaningful activities. This finding aligns with other studies suggesting that post-retirement work is associated with enhanced emotional well-being and greater satisfaction [34]. Additionally, the World Economic Forum reports that the global retirement age is expected to rise to accommodate an ageing population and boost productivity, with several European countries already leading the way in this shift [35].

In our study, sociocultural issues were primarily associated with family support, receiving respect, and opportunities for socializing. Beaudry et al. conducted a similar study among the older adult populations in India and Sri Lanka, emphasizing that family and community support are essential for long-term care. Many older adults, especially those managing disabilities or chronic illnesses, identified insufficient support as a primary factor leading them to seek residence in long-term care facilities. Additionally, these facilities largely depended on community assistance for daily operations due to minimal support from both family members and the government [36].

A Swedish study examining solutions for sociocultural issues among older adults highlighted how social media networking complemented traditional phone conversations, encouraging participation in social activities, events, and transportation support. Participants described engaging in community-oriented activities, such as walking a neighbor's dog, joining a church, cooking for others, or providing transportation, which fostered a sense of purpose and value [37]. Similarly, social support was seen as essential in this study; however, the participants in this study predominantly relied on their children or close family rather than community or neighborhood support for these interactions.

At the structural level, access to and utilization of healthcare services and medications remain a constant concern for older adults, as highlighted in our study. A recent study from India also reported that multi-morbidity significantly increases outpatient visits, hospitalization costs, and overall healthcare utilization, with affordability being a primary barrier to accessibility [38]. While cancer-related costs were especially burdensome in that study, the older adults in our study expressed more concern over managing conditions such as osteoporosis, hypertension, and diabetes.

A study from Malaysia identified several medication-related challenges among older adults, primarily involving administration, adherence, accessibility, and polypharmacy. It found that 44% of older adults skipped medications due to perceived ineffectiveness, and 60% experienced difficulties when changes occurred in medication color, size, or shape. Additionally, 15.5% reported trouble accessing information about medications from their primary physicians, and 62.3% felt they were on an excessive number of medications. Only 52.7% were aware of possible side effects. Dissatisfaction stemmed from long waiting times, insufficient guidance, and a lack of follow-up care [39]. In this study, participants similarly noted a need for family support in administering medicines and reported long hospital waiting times that led to missed follow-up visits. However, unlike the Malaysian study, this study's participants, both older adults and stakeholders, did not express concerns about polypharmacy, side effects, or difficulty accessing information. This difference may reflect the

limited health literacy or lower awareness of polypharmacy risks among participants from low socioeconomic backgrounds in our study.

Rajwar et al., in their scoping review protocol on income support programs in South Asian nations, highlighted several government initiatives aimed at reducing income insecurity among older adults. For example, India has established schemes such as the Indira Gandhi National Old Age Pension Scheme, the National Social Assistance Program, and the Senior Citizen Savings Scheme. Similarly, in Nepal, the government launched programs in 1994, including Old Age Allowances for all seniors over 60 years who are not covered by pensions and allowances for widowed, impoverished, or homeless older adult women. Bangladesh, with support from the World Bank, implemented similar programs over the past decade. Sri Lanka also has an Elderly Assistance Program, which provides cash transfers to older persons over 70 who lack income opportunities [40]. Notably, the review did not mention any comparable programs in Pakistan, although Pakistan does have pension systems and an Old Age Benefits Institution for government employees [41].

## Strengths and limitations

This study provides important evidence documenting ageing experiences across urban and rural settings in Pakistan, although several limitations warrant consideration. The inclusion of one urban and one rural district offered comparative depth but limited the generalizability of findings to other geographic settings in Pakistan. Pakistan's older population is very diverse, and perspectives across different provinces may reflect variation amongthe older population's preferences in terms of care. Additionally, recruitment through community mobilization may have excluded home-bound or socially isolated older adults, leading to potential under-representation of some of the most vulnerable groups.

The primary analysis was undertaken by a single researcher, which may have introduced interpretive bias and lack of inter-coder reliability. To enhance analytical rigor and accuracy, a second, qualitative researcher independently reviewed the coding framework, thematic structure, and supporting quotations to validate the analysis process. This peer review process, together with framework-guided coding, an audit trail, and debriefings with senior investigators, helped strengthen the rigor of findings. Translation from Sindhi and Urdu into English may also have led to slight loss of meaning despite multiple rounds of verification and transcript checking.

Despite these limitations, the study possesses several strengths. The study brought together perspectives from both older adults and stakeholders and used the socio-ecological model to examine ageing at multiple levels. In the rural setting, gender-segregated discussions ensured cultural sensitivity and gave participants the freedom to express their concerns more openly. Collectively, these features contribute to the methodological robustness and contextual relevance of the study findings, offering valuable evidence to inform interventions and practices supporting healthy ageing in LMICs.

## Conclusion

This study highlights the complexity of issues affecting ageing populations and emphasizes that these issues are not only personal but also deeply rooted in structural and sociocultural contexts. At the sociocultural level, addressing the lack of family support, respect, and avenues for socialization can be achieved by fostering community networks, promoting family time, and encouraging the delegation of responsibilities. On the individual level, physical and mental health concerns, as well as technological barriers, underscore the importance of personal preparation, gratitude, religious practices, and fostering a sense of importance. Overall, these findings emphasize the complexity of issues facing ageing populations and can be used to guide the creation of comprehensive plans and policies that support and empower healthy ageing.

## Supporting information

**S1 File. Study focus group discussion (FGD) guide.**
(PDF)

**S2 File. Condensed data analysis findings for older adults and stakeholders from Karachi and Thatta.**
(XLSX)

**S3 File. Thematic analysis findings for older adults and stakeholders in Karachi and Thatta.**
(PDF)

## Acknowledgments

The study team is thankful to the data collection team for their hard work.

## Author contributions

**Conceptualization:** Sonia Sameen, Maryam Pyar Ali Lakhdir, Justine Davies, Bilal Ahmed Usmani.

**Data curation:** Sonia Sameen, Javeria Mansoor, Maryam Pyar Ali Lakhdir, Muhammad Asim.

**Formal analysis:** Sonia Sameen, Javeria Mansoor, Muhammad Asim.

**Investigation:** Sonia Sameen, Javeria Mansoor, Maryam Pyar Ali Lakhdir.

**Methodology:** Sonia Sameen, Maryam Pyar Ali Lakhdir, Muhammad Asim, Rose Munyoki, Carolyn Greig, Justine Davies, Bilal Ahmed Usmani.

**Project administration:** Sonia Sameen, Maryam Pyar Ali Lakhdir, Bilal Ahmed Usmani.

**Resources:** Carolyn Greig, Justine Davies, Bilal Ahmed Usmani.

**Supervision:** Maryam Pyar Ali Lakhdir, Muhammad Asim, Carolyn Greig, Justine Davies, Bilal Ahmed Usmani.

**Visualization:** Sonia Sameen, Javeria Mansoor.

**Writing – original draft:** Sonia Sameen, Javeria Mansoor, Rose Munyoki.

**Writing – review & editing:** Sonia Sameen, Muhammad Asim, Carolyn Greig, Justine Davies, Bilal Ahmed Usmani.

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
