## [Decision Letter · Decision Letter 0]

8 Sep 2025

PGPH-D-25-01527

We Know What We Need: Older Adults’ and Stakeholders’ Perspectives on Ageing, Health, and Wellbeing in Pakistan

Dear Dr. Sameen,

Thank you for submitting your manuscript to PLOS Global Public Health. After careful consideration, we feel that it has merit but does not fully meet PLOS Global Public Health’s publication criteria as it currently stands. Therefore, we invite you to submit a revised version of the manuscript that addresses the points raised during the review process.

Please note that we have only been able to secure a single reviewer to assess your manuscript. We are issuing a decision on your manuscript at this point to prevent further delays in the evaluation of your manuscript. Please be aware that the editor who handles your revised manuscript might find it necessary to invite additional reviewers to assess this work once the revised manuscript is submitted. However, we will aim to proceed on the basis of this single review if possible.

Could you please carefully revise the manuscript to address all comments raised?

We look forward to receiving your revised manuscript.

Kind regards,

Helen Howard

Staff Editor

Journal Requirements:

1. In the online submission form, you indicated that All relevant data are included within the manuscript. Additional data are selectively available from the corresponding author upon reasonable request..

3. Uploaded as supplementary information.

Reviewers' comments:

Reviewer's Responses to Questions

**Comments to the Author**

1. Does this manuscript meet PLOS Global Public Health’s publication criteria?

Reviewer #1: Yes

2. Has the statistical analysis been performed appropriately and rigorously?

Reviewer #1: N/A

3. Have the authors made all data underlying the findings in their manuscript fully available (please refer to the Data Availability Statement at the start of the manuscript PDF file)?

Reviewer #1: No

4. Is the manuscript presented in an intelligible fashion and written in standard English?

Reviewer #1: Yes

Reviewer #1: Thank you for giving me the opportunity to review this manuscript which analyses a very relevant topic. Overall, the manuscript is well written and understandable to an external reader. However, I have a few comments that should be considered for a revision of the submitted manuscript:

- The introduction is well written and refers to the topic at hand. However, I would suggest a redesign of some phrases to improve the flow of the "story". The transition from the first paragraph to the second (line 83-84) comes very sudden and needs a sentence that transfers the focus from the challenges mentioned to the social determinants of health. The paragraph starting from line 95 deals with the reasoning for this article and I would put this rather towards the end as it is a perfect predecessor for the aim of the article. Overall, the introduction needs a bit more structure, alignment and logical flow rather than jumping from one topic to another.

- In the methods section, you briefly mentioned the broader multi-method research project, please also indicate the name (and any Grant Agreement Number or similar ID), so that interested readers can look it up. You also mentioned in the data collection to refer to the main study. Readers cannot do that without knowing the main study.

- Regarding the recruitment of older adults, it would be beneficial to state a few details on the recruitment process. How did the community mobilizer and research coordinator contact these people (lines 143-147)?

- Please also state any software or programs used (lines 178-179)

- You referred to the studies four main research questions (line 200) without mentioning them before. Please add them where appropriate (e.g. together with the study aim at the end of the introduction)

- I would also suggest to include a legend for table 1 (below). Readers may be able to proceed the bulk of information better, if barriers and solutions are placed in columns next to each other instead of rows. It makes it easier to compare and contrast (Column 1: Level of framework, Column 2: Themes Barriers, Column 3: Themes solutions)

- The figure is very appealing, but I am wondering what the message or added value behind it should be? It only focuses on the concerns and I think the value of the paper would be increased a lot to also highlight the solutions. Please think about redesigning the figure also integrating (and emphasizing) the solutions!

- The discussion offers an appropriate reflection of the results with contemporary literature. There are two citations missing: Mazumder et al. (lines 461-466) and Michigan Study (lines 494-500).

- What I am completely missing - which in my opinion is an absolute essential part of every scientific rigorous publication - is the critical reflection of the strengths and limitations of your work. Please include this in the discussion. E.g. a limitation coming to my mind is that only one reviewer conducted the analysis/coding (as mentioned in line 195), which may affect researcher bias and interpretive rigor, amongst others. Please critically go through your work and elaborate on the strengths and limitations of your methodology as well as a reflexivity part in the discussion!

- It would also be great to provide the semi-structured guide for the focus groups as supplementary material. Even the coding process would be very interesting for readers.

Thank you once more for giving me the opportunity to review this manuscript! I hope my comments are of help and I am looking forward to reading the revised version or even the published paper.

**Do you want your identity to be public for this peer review?** For information about this choice, including consent withdrawal, please see our Privacy Policy

Reviewer #1: No

---

## [Decision Letter · Decision Letter 1]

15 Jan 2026

We Know What We Need: Older Adults’ and Stakeholders’ Perspectives on Ageing, Health, and Wellbeing in Pakistan

PGPH-D-25-01527R1

Dear Dr Sameen,

We are pleased to inform you that your manuscript 'We Know What We Need: Older Adults’ and Stakeholders’ Perspectives on Ageing, Health, and Wellbeing in Pakistan' has been provisionally accepted for publication in PLOS Global Public Health.

Best regards,

Julia Robinson

Executive Editor

Reviewer Comments (if any, and for reference):

Reviewer's Responses to Questions

**Comments to the Author**

Reviewer #1: (No Response)

Reviewer #2: All comments have been addressed

publication criteria?

Reviewer #1: (No Response)

Reviewer #2: Yes

3. Has the statistical analysis been performed appropriately and rigorously?

Reviewer #1: (No Response)

Reviewer #2: N/A

4. Have the authors made all data underlying the findings in their manuscript fully available (please refer to the Data Availability Statement at the start of the manuscript PDF file)?

Reviewer #1: (No Response)

Reviewer #2: Yes

5. Is the manuscript presented in an intelligible fashion and written in standard English?

Reviewer #1: (No Response)

Reviewer #2: Yes

Reviewer #1: (No Response)

Reviewer #2: Thank you for writing this paper; it was a fascinating read.

The methodology is clearly described, well conceived, with an appropriate sampling approach and robust methods for data collection and analysis.

The study explores the impressions of older adults and other stakeholders regarding what it means to grow old in Pakistan, the barriers to healthy and successful ageing, and the areas where improvement may be possible.

The authors also summarise key gaps in the literature, situate their findings within the available global evidence, and offer a balanced analysis that is strengthened by important cultural context. This cultural framing meaningfully amplifies both the voices of the participants and the setting in which the data should be interpreted.

Overall, I support publication of this paper.

**Do you want your identity to be public for this peer review?** For information about this choice, including consent withdrawal, please see our Privacy Policy

Reviewer #1: No

Reviewer #2: No
